# Prospective cohort study to identify prevalence, risk factors and outcomes of infection associated kidney disease in a regional hospital in Malawi

Laura Isobel Carey ,[1,2,3] Sylvester Kaimba,[2] Saulos Nyirenda,[4] Karen Chetcuti,[5,6] Elizabeth Joekes,[1,6] Marc Yves Romain Henrion ,[1,2] Jamie Rylance[1,2]

[1]Department of Clinical Sciences, Liverpool School of Tropical Medicine, Liverpool, UK
[2]Malawi-Liverpool-Wellcome Trust Clinical Research Programme, Blantyre, Malawi
[3]Bristol Renal, Bristol Medical School, University of Bristol, Bristol, UK
[4]Department of Medicine, Zomba Central Hospital, Zomba, Malawi
[5]Kamuzu University of Health Sciences, Blantyre, Malawi
[6]Worldwide Radiology, Liverpool, UK

**Correspondence to**
Dr Laura Isobel Carey;
lcarey@mlw.mw

## ABSTRACT

**Objectives** Acute kidney injury (AKI) is a common and severe complication of community acquired infection, but data on impact in sub-Saharan Africa (SSA) are lacking. We determined prevalence, risk factors and outcomes of infection associated kidney disease in adults in Malawi.

**Design** A prospective cohort study of adults admitted to hospital with infection, from February 2021 to June 2021, collecting demographic, clinical, laboratory and ultrasonography data.

**Setting** Adults admitted to a regional hospital in Southern Region, Malawi.

**Primary and secondary outcome measures** The primary outcomes were prevalence of kidney disease and mortality by Cox proportional hazard model. AKI was defined according to Kidney Disease Improving Global Outcomes (KDIGO) guidelines. Secondary outcomes were risk factors for AKI identified by logistic regression and prevalence of chronic kidney disease at 3 months.

**Results** We recruited 101 patients presenting to hospital with infection. Median age was 38 years (IQR: 29–48 years), 88 had known HIV status of which 53 (60%) were living with HIV, and of these 42 (79%) were receiving antiretroviral therapy. AKI was present in 33/101 at baseline, of which 18/33 (55%) cases were severe (KDIGO stage 3). At 3 months, 28/94 (30%) participants had died, while 7/61 (11%) of survivors had chronic kidney disease. AKI was associated with older age (age: 60 years vs 40 years, OR: 3.88, 95% CI 1.82 to 16.64), and HIV positivity (OR: 4.08, 95% CI 1.28 to 15.67). Living with HIV was independently associated with death (HR: 3.97, 95% CI 1.07 to 14.69).

**Conclusions** Kidney disease is common among hospitalised adults with infection in Malawi, with significant kidney impairment identified at 3 months. Our study highlights the difficulty in diagnosing acute and chronic kidney disease, and the need for more accurate methods than creatinine based estimated glomerular filtration rate (eGFR) equations for populations in Africa. Patients with kidney impairment identified in hospital should be prioritised for follow-up.

## STRENGTHS AND LIMITATIONS OF THE STUDY

⇒ We provide data on prevalence, risk factors and consequences of acute kidney injury (AKI), which are scarce in sub-Saharan Africa (SSA).
⇒ We make use of the expanding availability of point of care ultrasound in Malawi and assess its role in the AKI assessment.
⇒ This is the first AKI study in Malawi reporting outcomes at 3 months, identifying significant kidney impairment and likely undiagnosed chronic kidney disease (CKD).
⇒ Diagnosis of AKI and the distinction between CKD is a challenge when baseline kidney function is unknown; pre-existing CKD may have been misclassified as AKI.
⇒ Our estimate of baseline kidney disease may be lower than actual prevalence, as eGFR equations and creatinine-based methods consistently underestimate kidney disease in SSA, making our findings even more serious and highlights the difficulties in using standard definitions of kidney impairment in this context.

deaths are in low-income and middle-income countries (LMICs).[1] Community acquired infections are a common cause of AKI in sub-Saharan Africa (SSA).[2] In patients presenting to hospital with AKI in SSA, 10%–70% have stage 3, requiring kidney replacement therapy, with an inpatient mortality of 44%.[3 4] The prevalence of AKI and CKD in community acquired infection in Malawi is unknown; no data exist for outcomes beyond hospital discharge.

The International Society of Nephrology initiative aims that 'no one should die of untreated AKI in low resource regions by 2025'.[5] In SSA, access to kidney function testing is limited, contributing to delayed AKI diagnosis and potential progression to kidney failure. In the region, patients are typically younger than those from high-income

## INTRODUCTION

Globally, 1.7 million people die from acute kidney injury (AKI) every year, 80% of

countries, and the majority will have no prior diagnosis or creatinine ascertainment. Available conventional estimators of baseline kidney function underestimate AKI incidence, for example, using the Modification of Diet in Renal Disease (MDRD) equation to back-calculate creatinine assuming an eGFR of $75 \, \text{mL/L/1.73 m}^2$ underestimates AKI incidence by>50%.[6]

Kidney ultrasound is standard practice in Malawi as part of the AKI assessment to identify chronic damage in the face of competition for dialysis beds. Patients with evidence of chronic kidney impairment are unlikely to be prioritised for kidney replacement therapy. In addition, kidney ultrasound may be used to identify those at increased risk of AKI, through assessment of kidney morphology, or by Doppler estimation of the Renal Resistive Index (RI), which directly correlates with intrarenal arterial resistance.

A systematic review of 176 patients found that raised RI values on ultrasound were associated with increased odds of persistent AKI.[7] Given the expanding availability of ultrasound in LMICs, we assessed RI and other sonographic parameters in the diagnosis and risk stratification of AKI in patients admitted to hospital with infection.

We aimed to estimate the prevalence of kidney disease at presentation and 3-month follow-up in non-hospital-acquired infection in Malawian adults. Second, we evaluated risk factors for kidney disease and outcomes of mortality.

## METHODS

### Setting

Zomba Central Hospital (ZCH) is a regional hospital in Southern Malawi serving both urban and rural populations with 60 adult medical inpatient beds. Healthcare is provided free at this government institution at the point of delivery. Approximately half of the patients usually present directly to the hospital and the other half are referred from primary or secondary care. Adults are triaged by a nurse and then reviewed by a doctor or clinical officer. If admission under a specialty team is deemed appropriate, they will be reviewed by an intern or registrar from the admitting specialty, and by a consultant usually within 24 hours.

Limited medical investigations are available on clinician request. Depending on availability, standard of care includes tuberculosis (TB) diagnostics (sputum smear microscopy, Xpert/MTB/Rif and urine lipoarabinomannan), malaria testing (usually rapid diagnostic test), plain film radiography, full blood counts and serum creatinine ascertainment. Inpatient HIV services include provider-initiated HIV testing and counselling, with CD4 count and viral load available on clinician request. A four-bed enhanced observation unit exists on the medical ward with provisions for oxygen therapy, but no cardiac monitoring or defibrillator. There is a five-bed intensive care unit, mostly used for surgical postoperative patients.

Local HIV prevalence is estimated at 17% among 15–49 year olds.[8]

At ZCH, we performed a prospective observational study recruiting adults (≥18 years) with suspected infection and a medical decision to admit to hospital. Suspected infection was deliberately broad to incorporate a wide spectrum of presenting physiology, and to be consistent with ongoing studies. This includes reporting in the last week any of: fever; rigours; night sweats and treatment with antibiotics. All patients in the admissions unit were approached for screening, Monday–Friday, 07:00–17:00.

Exclusion criteria were any of: people lacking capacity to consent, with no proxy consent available; pregnancy; non-severe uncomplicated localised infection with only local symptoms apparent on clinical examination by the screening clinical officer (such as cellulitis); detainees; trauma; admission to surgical or obstetric/gynaecological ward and previously recruited to the study. Responsibility for care and treatment decisions remained with the usual clinical team. Participants were followed-up as inpatients at 48 hours, and at 3 months in the community.

### Sampling and laboratory methods

Patients provided blood and urine at baseline, 48 hours and 3 months. Creatinine, blood urea nitrogen, glucose, ionised calcium, sodium, potassium, chloride and bicarbonate were assessed by iSTAT CHEM8 cartridges (Abbott Point of Care, Princeton, NJ). The handheld iSTAT system measures creatinine using an enzymatic assay traceable to the US NIST Standard Reference material SRM909 with a reportable range of 0.20–20.0 mg/dL. Standard of care for measurement of serum creatinine in ZCH is using a sarcosine oxidase (enzymatic) method on a Mindray BS360-E machine. Often this test is unavailable due to reagent stock outs which are supplied by the government.

Point of care HIV testing was done for those with unknown status or no recent negative test. Urine was tested by dipstick for blood and protein, and positive samples further quantified using a laboratory analyser (Mindray BS360).

### Ultrasound

Point of care ultrasound assessed kidney size, corticomedullary differentiation, echogenicity and the presence of any hydronephrosis defined as a dilated appearance of the collecting system. Assessment was made by a clinical officer who underwent dedicated training, including direct supervision by a consultant radiologist before study commencement.

RI was measured at each pole of each kidney (upper, middle and lower) and an average taken across poles and between kidneys giving one RI value per participant. A portable Mindray DP-50 machine with 5MHz convex probe was used throughout. The assessor was blinded to the kidney diagnosis.

## AKI assessment

Creatinine was measured on admission, at 48-hour and at 3-month follow-up.

AKI was defined according to Kidney Disease Improving Global Outcomes (KDIGO) as a 1.5-fold increase in baseline creatinine from the estimated baseline.[9] AKI severity as fold increase over estimated baseline was defined according to KDIGO as: stage 1 (1.5–1.9×); stage 2 (2.0–2.9×) and stage 3 (≥3.0×).

We refer to incident AKI at 48 hours if AKI criteria were not met at baseline, but there was a creatinine rise within 48 hours. AKI recovery was defined as a decrease of 0.3 mg/dL between baseline and 48 hours or a normal creatinine (<1.3 mg/dL) at 3 months.

Acute kidney disease (AKD) was assessed on admission and at 3 months and defined according to KDIGO as: under 3 month duration, glomerular filtration rate (GFR) <60 mL/min/1.73 m$^2$ or decrease in GFR by ≥35% or increase in creatinine by>50% and/or albuminuria/ haematuria.[10] Urine output was not assessed as it is not universal clinical practice in Malawi.

AKI associated with infection, particularly in LMICs, affects patients unlikely to have a documented baseline creatinine, necessitating estimation through back-calculation.[6] No participants had a previous documented creatinine result. Given the young age distribution of the population with relatively few comorbidities, baseline creatinine was estimated using an MDRD equation, assuming a GFR of 100 mL/min.[6] MDRD was modified to exclude the race coefficient as eGFR equations using ethnicity adjustment factors overestimate GFR, leading to underdiagnosis of CKD and negative bias in people of black ethnicity.[11–13]

The full-age spectrum equation was used to back calculate an estimate of baseline creatinine: using MDRD may be imprecise in estimating baseline creatinine in older adults, resulting in lower baseline estimates and more AKI diagnosis at older ages.[14] In addition, we estimated AKI using lowest creatinine during admission as a baseline value. A sensitivity analysis compared the effect of each method of baseline creatinine estimation on AKI incidence.

## Illness severity

Illness severity was measured by quick Sepsis-related Organ Failure Assessment (qSOFA) score, which uses three criteria: one point for low blood pressure (systolic blood pressure ≤100 mm Hg), high respiratory rate (≥22 breaths/min) or altered mental state (Glasgow Coma Scale <15).

## Mortality and CKD

Mortality was defined as death by 48 hours or 3 months. CKD was assessed at 3 months and defined as an eGFR of <60 mL/min/1.73 m$^2$ calculated using combined results from the Chronic Kidney Disease Epidemiology Collaboration (CKD-EPI) and the 4-parameter MDRD study equation, for a non-IDMS (isotope-dilution mass spectrometry)-traceable creatinine method, without race adjustment.[15–17] GFR estimates were reported as adjusted for body surface area.[15] Those with an eGFR >60 mL/min/1.73 m$^2$ at presentation were excluded from the CKD definition as the 3-month duration had not been proven. Those with an eGFR <60 mL/min/1.73 m$^2$ at presentation and 3-month follow-up were given a diagnosis of CKD.

## Statistical analysis

Statistical analyses were performed in R (V.4.0.2).[16 18] Data and code are publicly available in a GitHub repository.[19] Summary statistics were calculated for the cohort, which was then divided into AKI and no-AKI groups, described using median and IQR for continuous variables and proportions for categorical variables. Two sample t-tests or non-parametric tests, depending on data distribution, were used to compare variables between groups, and Fisher's test for proportions. All methods and reporting follow the Strengthening and Reporting of Observational Studies in Epidemiology guidelines.[20]

A directed acyclic graph was used to examine the hypothesised causal relationship between HIV exposure and AKI (online supplemental figure 1), and to identify any possible confounding. Age was identified as a confounder, and admission to hospital a collider, as participants were conditioned on this variable by inclusion in the study. Risk factors for AKI were considered and used for logistic regression analysis. Logistic regression outputs were reported as OR and 95% CIs.

Age was included in the adjusted model as a cubic spline term with three knots, meaning that we used age as a continuous variable rather than categorising it into a limited set of age bands. The effect of age on probability of AKI was modelled using predicted probabilities from the adjusted model (when qSOFA score=2, HIV status=positive and diastolic blood pressure=75 mm Hg). ORs for different ages were generated by comparing odds to the reference age, chosen here to be 40 years.[21] Bootstrap sampling and the percentile method with 1000 replicates were then used to construct 95% CIs for the OR.

Kaplan-Meier survival analysis was conducted using Cox proportional hazards adjusted for age and qSOFA score.

## Sample size

A power calculation for specific risk factors was not performed before initiation of this exploratory study. However, a post-hoc power calculation was performed in G Power (V.3.1), given the achieved sample size (88 participants with recorded HIV and AKI status), using HIV as the risk factor.[22] Given the observed prevalence of AKI in HIV negative participants at 23% (8/35), with HIV prevalence at 60% (53/88), for a two-tailed test, with 80% power and 0.05 alpha, the study was powered to detect effect sizes with an OR at least as large as 3.79.

**Table 1** Characteristics of participants

| Variable | Value |
|---|---|
| **Demographics** | |
| Age (years) | 38 (29–48) |
| Male sex | 47/101 (47) |
| Body mass index (kg/m) | 22(19–23) |
| **HIV/TB/Malaria** | |
| HIV infected*, n (%) | 53/88 (60%) |
| Receiving antiretroviral therapy, n (%) | 42/53 (79%) |
| Receiving co-trimoxazole preventative therapy, n (%) | 40/53 (75%) |
| Receiving isoniazid preventative therapy, n (%) | 11/53 (21%) |
| History of receiving TB treatment, n (%) | 11/101 (11%) |
| Prior malaria, n (%) | 21/101 (21%) |
| **Comorbidities** | |
| Pre-existing diagnosis of diabetes mellitus, n (%) | 1/101 (1%) |
| Pre-existing diagnosis of hypertension, n (%) | 7/101 (7%) |
| **Drugs** | |
| Antibiotic use prior, n (%) | 80/101 (79%) |
| Traditional/over the counter medications, n (%) | 4/101 (4%) |
| **Symptoms** | |
| Vomiting, n (%) | 25/101 (25%) |
| Diarrhoea, n (%) | 16/101 (15%) |
| Cough, n (%) | 43/101 (43%) |

Values are median (IQR).
*HIV status missing for 16 participants.

## Missing data

Missing data were less than 18% across all variables (online supplemental figure 2), so results of a complete case analysis are reported. The main reason for missing follow-up data was participants being uncontactable, or unable to get to ZCH for a follow-up blood test due to transport or logistical reasons.

## Patient and public involvement

At the end of the study period, participants, patients and staff at ZCH were invited to a kidney awareness day, where key results were disseminated using simple leaflets, presentations and a group discussion.

## RESULTS

Between 3 February 2021 and 19 July 2021, 101 eligible patients admitted with infection were recruited. The study flow diagram is summarised in online supplemental figure 3. Table 1 summarises the baseline characteristics of the participants. There were 47/101 (47%) male participants. The median age was 38 years (IQR: 29–48 years). Median follow-up time was 92 days (IQR: 71–95 days). Median number of days unwell prior to hospital presentation was 7 days (IQR: 3–21 days).

For those who HIV status was known (88/101), 53/88 (60%) of participants were people living with HIV. Among people living with HIV, 42/53 (79%) were receiving antiretroviral therapy, 40/53 (75%) co-trimoxazole

preventive therapy and 11/53 (21%) isoniazid preventive therapy. Among all participants recruited, 11/101 (11%) had a history of receiving treatment for tuberculosis and 21/101 (21%) for malaria within the past month. The majority, 80/101 (79%), reported antibiotic use prior to presentation and 4/101 (4%) reported using over the counter medications or traditional medicines. No participants had a known prior diagnosis of CKD, 1/101 (1%) reported a history of diabetes, and 7/101 (7%) a history of hypertension.

A histogram of baseline creatinine values in mg/dL is presented in online supplemental figure 4. Ultrasound, laboratory and derived indices are summarised in online supplemental table 1. Participants were classified according to AKI status. Using assumed GFR of 75 mL/min and 100 mL/min to estimate baseline creatinine, 28 (28%) and 33 (33%), respectively, met the definition for AKI at presentation. Back calculation of baseline creatinine using the full age spectrum equation identified 36/101 (36%) cases of AKI, whereas using lowest creatinine value identified 17/101 (17%) cases of AKI. Overall, the majority of patients (63/101, 62%) met our definition of AKD at presentation.

Outcomes, and the effect of different methods to estimate baseline creatinine on AKI prevalence, are presented in table 2. The prevalence of CKD and AKD among survivors with creatinine values available at 3 months was 7/61 (11%) and 11/61 (18%), respectively. Mortality among those contactable at 3 months (94/101) was 28/94 (30%).

Demographic univariable associations with AKI status are summarised in online supplemental table 2. Ultrasound and urine-related univariable associations are presented in online supplemental table 3. There was no significant difference in kidney appearances on ultrasound and RI between participants with AKI compared with those without AKI. Observation and laboratory univariable associations are in online supplemental table 4 and the effect of age on AKI when the full age spectrum equation is used to back calculate creatinine is presented in online supplemental table 5. Online supplemental table 6 shows the effect of each definition of AKI and AKD on the mortality outcome: death by 3 months.

Multivariable associations with AKI status (based on estimated baseline creatinine assuming GFR: 100 mL/min) identified by logistic regression are presented as unadjusted and adjusted ORs and 95% CIs in table 3.

Increasing age, living with HIV, microscopic haematuria and proteinuria were most associated with AKI on logistic regression analysis. Online supplemental figure 5 shows the odds of AKI for different ages, qSOFA scores and HIV status, with systolic blood pressure held constant at 80 mm Hg. Online supplemental figure 6 presents the effect of age on probability of AKI, and online supplemental figure 7 presents the RI distributions by AKI status as a boxplot.

Time-to-event analysis, adjusted for age and qSOFA score demonstrated an increased hazard of death for HIV

**Table 2** Outcomes

| Timepoint | Value |
|---|---|
| **0 hours** | |
| Creatinine (mg/dL) | 0.80 (0.60–1.60) |
| eGFR (mL/min/1.73 m²) MDRD | 93(41–141) |
| Creatinine clearance (mL/min) (Cockroft and Gault) | 94(35–129) |
| AKI (assumed GFR 100 mL/min) n (%) | 33/101 (33%) |
| AKI (assumed GFR 75 mL/min), n (%) | 28/101 (28%) |
| AKI (full age spectrum equation), n (%) | 36/101 (36%) |
| AKI using lowest creatinine as baseline, n (%) | 17/101 (17%) |
| AKD, n (%) | 63/101 (62%) |
| Stage 3 AKI, n (%) | 18/33 (55%) |
| Stage 2 AKI, n (%) | 8/33 (24%) |
| Stage 1 AKI, n (%) | 7/33 (21%) |
| **48 hours** | |
| Incident AKI*, n (%) | 3/80 (4%) |
| Recovered AKI, n (%) | 18/33 (55%) |
| Recovered AKD, n (%) | 38/63 (60%) |
| Alive at 48 hours†, n (%) | 91/94 (97%) |
| **3 months** | |
| Alive at 3 months†, n (%) | 66/94 (70%) |
| AKD‡, n (%) | 11/61 (18%) |
| CKD (eGFR <60 mL/min/1.73 m² at baseline and 3 months)‡, n (%) | 7/61 (11%) |

Values are median (IQR).
*48-hour creatinine missing n = 21.
†Lost to follow-up n = 7.
‡3-month creatinine missing in survivors n = 5.
AKD, acute kidney disease; AKI, acute kidney injury; CKD, chronic kidney disease; eGFR, estimated glomerular filtration rate; GFR, glomerular filtration rate; MDRD, Modification of Diet in Renal Disease.

positive participants (HR: 3.97, 95% CI 1.07 to 14.69) (figure 1, Kaplan-Meier curves).

## DISCUSSION

Adults presenting with infection in Zomba, Malawi are young, and predominantly living with HIV. Prevalence of AKI depended on the method of defining baseline creatinine, but was between 17% and 36%, with that of AKD at 62%. The majority of AKI was severe (stage 3), and mortality at 3 months was 28/94 (30%). Living with HIV and increasing age were risk factors for presentation to hospital with AKI, and there was a significant prevalence of kidney disease at 3 months (11% CKD and 18% AKD). Living with HIV was independently associated with death at 3 months.

Other limited studies examining kidney impairment in Blantyre, Malawi found 21% of medical admissions had evidence of kidney disease on admission (17% AKI and 3% AKD), which is lower than our reported prevalence for both, particularly AKD.[4] Of the Blantyre participants with AKI, 60% were stage 3 AKI, and inpatient mortality was 44% in patients with AKI compared with 14% with no kidney disease.[4] The mortality in patients with AKI in Blantyre is likely higher than ours in Zomba (44%

vs 30%), as it included AKI in the context of general medical problems: malignancy, liver disease, heart failure and stroke, rather than AKI in the context of infection.

Worldwide, the incidence of AKI is not well known. One report of 312 studies, including >3.5 million patients mostly from hospitals in high-income countries, suggests a pooled incidence of 22% (95% CI 19 to 24) and a mortality rate of 23%.[23] In high-income country intensive care settings, dialysis requiring AKI mortality rates can vary between 40% and 60%.[24–26] Organ dysfunction was not a requirement for inclusion in our study, which may explain why our mortality is relatively lower.

To the best of our knowledge, this is the first AKI study in Malawi to obtain creatinine values at 3 months post discharge. Our data suggest a CKD prevalence of 7/61 (11%). How our data compares to the true community population prevalence of CKD in Malawi is not known. Normative ranges for GFR have not been established in African populations, including whether a cut-off of GFR of less than 60 mL/min/1.73 m² is an appropriate definition of CKD.[27]

In a recently published cohort, the African Research on Kidney Disease (ARK) study, in populations from Malawi, Uganda and South Africa, the prevalence of measured GFR<60 mL/min/1.73 m² using iohexol plasma clearance was 19%. This is nearer our estimate of AKD prevalence at 3 months, than our estimate of CKD.[27] A cross-sectional survey of urban and rural Malawi using the CKD-EPI equation suggested a prevalence of eGFR <60 mL/min/1.73 m² of 1%, and a systematic review of 13 countries in SSA estimated the pooled prevalence of CKD to be 14%.[28 29]

Prevalence reported by eGFR equations rather than measured GFR are likely to be significant underestimates, as highlighted by the important findings from the ARK study. Estimating GFR using serum creatinine substantially underestimates the population level burden of impaired kidney function in Africa.[27] AKD assessment, which is broader and includes albuminuria and haematuria, may be more representative of the true burden. Regardless, we urgently need more accurate methods to report kidney disease for the region, or risk continued underreporting.

An additional complication is that patients presenting with infections in Malawi do not have documented baseline creatinine values, which can be addressed by back-calculation estimation by GFR equations. However, the optimal equation to accurately estimate GFR in Malawi (and across SSA) is uncertain. In studies from SSA, coefficients for African–American ethnicity consistently overestimate GFR.[30] Similarly, applying creatinine-based eGFR formulae in HIV without adjusting for BMI tends to overestimate GFR and, therefore, underestimate CKD burden.[12]

We compared the different equations for estimating baseline creatinine and prevalence of AKI. Using lowest creatine during admission as a baseline identified fewer cases of AKI (17%) than the MDRD and full age spectrum

**Table 3** Logistic regression analysis of risk factors associated with AKI

| ORs and 95% CIs for AKI | | | |
|---|---|---|---|
| | | **AKI OR (95% CI)** | |
| **Characteristic** | **Category** | **Unadjusted** | **Adjusted for age and HIV status** |
| Age | 20 | 1.07 (0.25 to 4.26) | 1.81 (0.86 to 22.19) |
| | 30 | 0.91 (0.52 to 1.61) | 0.91 (0.66 to 2.18) |
| | 40 (ref.) | 1 | 1 |
| | 50 | 1.63 (1.14 to 2.21) | 1.59 (1.14 to 2.46) |
| | 60 | 3.45 (1.60 to 7.20) | 3.88 (1.84 to 16.64) |
| | 70 | 8.07 (2.24 to 30.72) | 14.59 (3.96 to 278.02) |
| | 80 | 18.93 (3.14 to 137.70) | 62.02 (9.82 to 5185.71) |
| Sex | Female (ref) | 1 | 1 |
| | Male | 0.96 (0.55 to 1.68) | 1.04 (0.38 to 2.82) |
| BMI (kg/m$^2$) | | 0.97 (0.90 to 1.04) | 0.92 (0.79 to 1.05) |
| HIV | Positive | 1.65 (0.82 to 3.32) | 4.08 (1.28 to 15.67) |
| | Negative | 1 | 1 |
| Comorbidity | Hypertension | 2.40 (1.37 to 4.21) | 3.19 (0.43 to 29.27) |
| Structural | Microscopic haematuria | 3.92 (1.77 to 8.66) | 6.34 (1.99 to 24.87) |
| | Proteinuria | 2.98 (1.75 to 5.07) | 4.05 (1.39 to 12.22) |
| | Kidney size | 0.86 (0.68 to 1.09) | 1.32 (0.82 to 2.15) |
| | Increased echogenicity | 1.33 (0.77 to 2.32) | 1.61 (0.59 to 4.44) |
| | Loss of corticomedullary differentiation | 1.20 (0.69 to 2.09) | 0.99 (0.36 to 2.73) |
| | Resistive index | 58.48 (1.32 to 2596.32) | 70.11 (0.79 to 84 185.95) |
| Physiological | qSOFA score | 1.31 (0.81 to 2.11) | 1.79 (00.76 to 4.45) |
| | Oxygen saturations | 0.91 (0.77 to 1.00) | 0.98 (0.81 to 1.21) |
| | Diastolic blood pressure* | 1.01 (1.01 to 1.02) | 1.02 (0.99 to 1.05) |
| | Bicarbonate | 0.78 (0.69 to 0.86) | 0.79 (0.68 to 0.90) |

Parameter estimates expressed as unadjusted and adjusted ORs and 95% CIs.
*Higher diastolic blood pressures were associated with AKI in the unadjusted model.
AKI, acute kidney injury; BMI, body mass index; qSOFA, quick Sepsis-related Organ Failure Assessment.

back calculation equations (28%–36%), but may have been less likely to misclassify CKD as AKI. Comparatively, AKD encompasses a much broader definition and identified many more cases of kidney impairment (62%). Mortality at 3 months was similar for each AKI definition except lowest creatinine as baseline, for which there were no deaths, likely because this definition excluded participants with the most severe but unchanged kidney impairment.

In this context, the inherent limitations of creatinine based AKI definitions are likely to substantially underestimate actual prevalence. Despite using high-income country based 'normal' eGFR values, this study still shows a high proportion of likely baseline kidney disease. Even if a substantial proportion of the baseline AKI was in fact undiagnosed CKD, this still represents a significant burden of kidney disease in adults in Malawi who present to hospital with infections. Furthermore, the binary AKI/ CKD definitions are difficult, and dichotomisation may not even be meaningful or helpful in this context.

We did not find ultrasound to be useful in identifying patients at risk of AKI in our cohort. Furthermore, resistive indices can be technically challenging to obtain and are not commonly practiced in Malawi, and we note the wide CI around our OR estimates. Our data does not support use of the RI in the AKI assessment and further investigation, with larger datasets, is needed. We suggest point-of-care ultrasound use as part of the AKI assessment, but not as a surrogate for serum creatinine measurement.

Most deaths occurred after discharge rather than in hospital, indicating long-term mortality associated with hospitalisation with infection. Combined with the high prevalence of kidney disease identified at 3 months, community follow-up of patients is essential. To reduce post-discharge deaths, patients should be counselled on avoiding nephrotoxic medications which are commonly used (eg, non-steroidal anti-inflammatories), and advice given on how to prevent future episodes of AKD. Due to limited resources, follow-up should be prioritised for those identified with kidney impairment. Outpatient

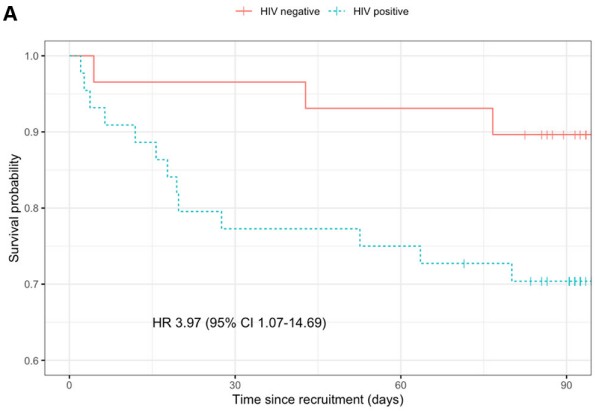

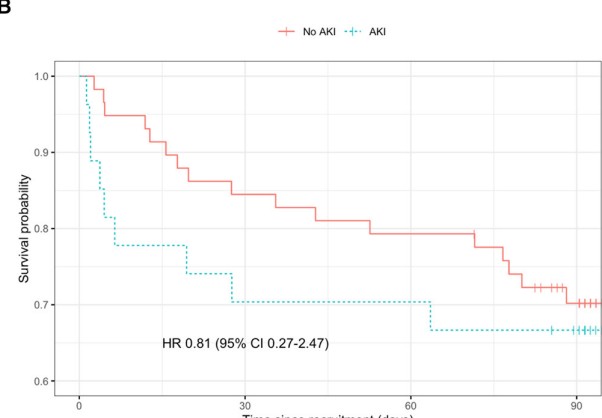

**Figure 1** Kaplan-Meier estimate of survival function following hospital admission with infection. (A) HR for death according to HIV status from Cox proportional hazards model adjusted for age and qSOFA score with 95% CIs. (B) HR for death according to AKI status from Cox proportional hazards model adjusted for age and qSOFA score with 95% CIs. AKI, acute kidney injury; qSOFA, quick Sepsis-related Organ Failure Assessment.

review after discharge should include assessment for recovery of kidney impairment, new AKD or formal diagnosis of CKD. Those identified with AKD/CKD will need ongoing follow-up for CKD assessment and management to prevent progression to kidney failure.

Our study has several limitations. We do not know the true prevalence of CKD at presentation, given there were no historical serum creatinine results and patients did not report known CKD. This may have resulted in misclassification of CKD as AKI and AKD. It is reasonable to assume that the 7 cases of CKD (eGFR <60 mL/min/1.73 m$^2$ at both presentation and 3-month follow-up) were misclassified as both AKD and AKI at presentation. Evidence of recovery at 3 months was seen in 18/33 (55%) of those with AKI and 38/63 (60%) with AKD, suggesting true AKI/AKD rather than misclassified CKD.

We were unable to identify incident CKD in those who had normal kidney function (eGFR >60 mL/min/1.73 m$^2$) at presentation at follow-up as the 3-month rule would not be observed. However, there was frequent AKD at 3 months, 18% new kidney impairment in patients who

had normal kidney function on admission. This study also highlights the significant prevalence of undiagnosed CKD (eGFR <60 mL/min/1.73 m$^2$ at both presentation and 3-month follow-up), of which none of the participants had prior knowledge. The high prevalence of HIV in the cohort increases the likelihood of baseline kidney disease, and indeed the majority of the CKD at 3 months was in people living with HIV (5/7 (71%)). Estimates for CKD prevalence in people living with HIV across Africa are between 1% and 46% with a pooled prevalence of 6%.[31]

The other significant limitation is related to there being no validated method to estimate GFR and baseline creatinine for Africans, in health or acute illness. We estimated baseline creatinine using an assumed eGFR of 100 mL/min, which resulted in a 5% difference in AKI prevalence compared with an assumed eGFR of 75 mL/min. An assumed eGFR between 75 mL/min and 100 mL/min is supported by the median (IQR) iohexol measured GFR in Malawians reported in the ARK study: 86 mL/min/1.73 m$^2$ (72–100 mL/min/1.73 m$^2$) and 92 mL/min/1.73 m$^2$ (77–104 mL/min/1.73 m$^2$) for women and men, respectively.[27]

We used MDRD and CKD-EPI equations to report eGFR normalised to 1.73 m$^2$ body surface area; however, adults in Malawi may have a smaller body surface area. The median (IQR) body surface area for Malawian females and males in the ARK study was 1.70 m$^2$ (1.50–1.80 m$^2$) and 1.70 m$^2$ (1.60–1.80 m$^2$), respectively.[27] Urine output was not used to define AKI, which may have led to underestimation and imprecision of AKI staging. Partial pressure of carbon dioxide (PCO$_2$) was not measured, preventing correction of iSTAT measurements for PCO$_2$. It was not possible to ascertain causes of death, and our study was not powered to detect moderate or small associations.

To prevent deaths from untreated AKI in low-resource regions by 2025, diagnosis of AKI and kidney disease needs to be appropriate for the local context. We urgently need more accurate methods to assess kidney function for specific African populations. eGFR and creatinine-based equations underreport the burden of kidney disease and the distinction between AKI and CKD is difficult.[1] Interventions to reduce the impact of kidney disease in Malawi should focus resources on identifying patients with kidney impairment for follow-up to prevent progression to kidney failure. This is essential given the scarcity of kidney replacement therapy.

**Contributors** LIC and JR conceived and designed the study. SN was involved in the conceptualisation and project administration. EJ and KC were involved in conceptualisation and point of care ultrasound. KC delivered the ultrasound training. SK collected the data and performed the ultrasounds. LIC analysed the data and wrote the manuscript and is responsible for the overall content as the guarantor. MH advised on statistics. All authors read, commented on and approved the final manuscript.

**Funding** LIC was funded by an institutional translational partnership award (grant: 219633/Z/19/Z) from the Wellcome Trust to Malawi-Liverpool-Wellcome Trust Clinical Research Programme (MLW). JR is funded by a Wellcome Career

Development Fellowship (grant: 206545/Z/17/Z) and by the National Institute for Health Research (NIHR) (17/63/42) using UK aid from the UK Government to support global health research. The views expressed are those of the authors and not necessarily those of the National Health Service, NIHR or the Department of Health and Social Care. For the purpose of open access, the author has applied a CC BY public copyright licence to any Author Accepted Manuscript version arising from this submission.

**Competing interests** None declared.

**Patient and public involvement** Patients and/or the public were involved in the design, or conduct, or reporting, or dissemination plans of this research. Refer to the Methods section for further details.

**Patient consent for publication** Not applicable.

**Ethics approval** This study involves human participants and was approved by College of Medicine Research Ethics Committee, University of Malawi (P.03/19/2625) and the Liverpool School of Tropical Medicine Ethics Committee (18-062). Participants gave informed consent to participate in the study before taking part.

**Provenance and peer review** Not commissioned; externally peer reviewed.

**Data availability statement** Data are available in a public, open access repository. Data and code available at https://github.com/careyla/PARIS.

**ORCID iDs**
Laura Isobel Carey http://orcid.org/0000-0003-4158-4527
Marc Yves Romain Henrion http://orcid.org/0000-0003-1242-839X

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
