## [Reviewer comments · BMJ Open]

ARTICLE DETAILS

TITLE (PROVISIONAL)	Prospective cohort study to identify prevalence, risk factors and outcomes of infection associated kidney disease in a regional hospital in Malawi.
AUTHORS	Carey, Laura; Kaimba, Sylvester; Nyirenda, Saulos; Chetcuti, Karen; Joeke, Elizabeth; Henrion, Marc; Rylance, Jamie

VERSION 1 – REVIEW

REVIEWER	Evans, Rhys University of Malawi College of Medicine
REVIEW RETURNED	18-Jul-2022

GENERAL COMMENTS	Thank you for giving me the opportunity to review this manuscript. Carey et al undertake a prospective study to investigate the prevalence and outcomes of sepsis/infection related AKI in a secondary/tertiary hospital in Malawi. They identify risk factors for the development of AKI and AKI outcomes; and they also undertake an exploratory analysis of the use of kidney USS in AKI risk stratification. 101 patients are included in a cohort of young patients with a high prevalence of HIV, typical for hospitals in this region. AKI prevalence is 33% with the majority AKI Stage 3 – similar to previous studies of AKI in Malawi. Mortality at 3 months in the entire cohort was 35% with persistent kidney disease present in 20% of those followed up. AKI development associated with SOFA score, age, and HIV positivity. HIV associated with death. The authors should be congratulated for undertaking a rigorous study in a challenging resource poor setting. The data complements previous epidemiological studies of AKI undertaken in Malawi and the region. In my mind, its novelty lies in the follow up data collected both in terms of patient and kidney outcomes (expanded on below) and in the exploratory use of kidney USS, albeit the rationale for undertaking this analysis should be clarified, as below. Comments / suggestions: 1. Whilst some readers of this manuscript will have an insight in to the setting / resources available where the study was undertaken this is unlikely to be universal. It may be helpful to provide a brief overview of resource limitations (and availability) in the setting where the study was conducted so patient outcomes etc can be interpreted in this light. I.e. what diagnostics /treatments are routinely available. What monitoring / lack therefore. Do patients present directly to the hospital / are they referred etc. How is healthcare set up in Malawi (funding etc).
--

	2. the rationale for the investigation of the use of bedside kidney USS I feel needs expanding. There are no references to previous use of renal USS in the risk stratification of AKI,; for example Is there previous evidence (low or high resource setting) to suggest kidney USS is of use in risk stratifying AKI? Whilst use of USS is routine in the diagnosis of obstruction or alteration in renal arterial / venous flow, I don't feel it is used much beyond this in AKI management as part of standard clinical care. 3. Inclusion criteria: the authors state that this is a study of AKI in the setting of 'sepsis', which as in the introduction they state refers to 'life threatening organ dysfunction triggered by a dysregulated host response to infection'. However the inclusion criteria includes patients with 'suspected infection that are admitted to hospital'. Do the authors feel that all of these patients have 'sepsis' and if so it would be useful to outline the definition of sepsis they have used. Would it be more accurate to reframe the study to talk about 'infection related AKI' as opposed to 'sepsis related'. They go on to outline that non-severe infections were excluded. How was the severity of infection determined and by whom? 4. Definitions of AKI: the authors allude to the challenges of AKI diagnosis in resource limited settings in particular given the almost universal lack of historical creatinines through which to determine a baseline value. They use an imputed baseline back calculated based on an assumed GFR of 75 and 100 using MDRD. Did any of the patients have a previous creatinine available or was back calculation used in all patients? As well as creatinines prior to admission, did the authors use / consider use of the lowest creatinine during admission as a baseline value? Another approach the authors may want to consider in the paper is the concept of acute kidney disease. This has become an increasingly used term that I feel is particularly useful in low resource settings when there is a lack of a baseline value (see Lameire et al, KI, 2021: https://www.kidney-international.org/article/S0085-2538(21)00662-1/fulltext). AKD can be associated with or without AKI, and allows harmonization with CKD diagnosis. In determining GFR for the purposes of CKD (and hence potentially AKD) definition I note the authors used MDRD - What were reasons for this and was CKD-EPI (race unadjusted) considered? Are there differences in CKD prevalence when different estimating equations are used? What is the prevalence of AKD in the population described – ie. can more patients with kidney disease be identified using this definition? 5. the authors recruited patients from a 5 month period (jan-july). They may want to comment on malaria seasonality of this period and how this may affect generalizability of results. 6. what are the infections that led to 'sepsis' in the cohort described. How were these infections diagnosed (eg. did all patients with malaria have an film / antigen test)? 7. Did any of the patients need / receive renal replacement therapy? 8. A real strength of the paper I feel is the completeness of the follow up data and the authors should consider expanding more on
--	--

	this. They give the mortality rates at 48 hours and at 3 months. Are there differences in outcomes (mortality) according to kidney disease classification at presentation? If not it may still be useful to present this data. Moreover, in the discussion they outline that most deaths occurred post discharge. It may be good to highlight this in the results – ie. what was median LOS – what proportion deaths occurred post discharge. What are the implications for clinical practice of these findings? In addition, what was the kidney classification of patients with CKD at 3 months on admission. I assume that the patients with 'ckd' had a GFR <60 on initial presentation in order for the CKD definition to be made. They may want to discuss the possibilities of these patients having undiagnosed CKD (at presentation), or de novo CKD. Did any patients have 'normal kidney function' on presentation and kidney disease at 3 months? – these patients cant be classified as CKD as 3 month duration has not been proven – its unclear if any such patients have been included in the 'ckd group'. Again the authors may want to consider the concept of 'kidney disease' at this timepoint prior to determining whether this is confirmed to be chronic (otherwise acute). Lastly, given 1/5 of patients assessed had kidney disease at the 3 month period it should be emphasized the importance of follow up of this group. 9. The renal USS analysis I feel is overstated. None of the parameters they refer to reach statistical significance (univariate). It would be interesting to know what the definitions were of: hydronephrosis (21% in AKI and 13% in no AKI seems high to me); loss of CM differentiation; increased echogenicity. A lot of these parameters are somewhat subjective. Was the operator blinded to the kidney diagnosis? The RI is the least subjective in my mind and there are striking similarities in this between the groups (albeit note multivariate findings) 10. the authors comment that the CKD prevalence is likely to be low. This statement I feel should be made with caution given the risk factors present in the cohort (in particular the prevalence of HIV) 11. the primary outcomes of the study were to determine the prevalence of AKI and their outcomes. However, in the discussion, there is relatively little comment on these. How did the prevalence and outcomes (kidney / patient) compare to other studies in malawi and the region? How do they compare to high resource settings? Was anything different found in this cohort to what has been previously shown and what might be the reasons for any differences found? 12. As above, how should the finding of high post discharge mortality and development of CKD (/persistent CKD) inform clinical practice in the region. Ie. should patients be monitored more closely post discharge and what are the resource restrictions of this? 13. the study ref in 23 has recently been published. The authors may want to update and discuss implications of these findings for this study. https://www.thelancet.com/journals/langlo/article/PIIS2214-109X(22)00239-X/fulltext
--	---

	Thank you again and well done again to all the authors for their work so far.
--	---

REVIEWER	Conroy, Andrea Indiana University School of Medicine
REVIEW RETURNED	21-Jul-2022

GENERAL COMMENTS	Thank you for the opportunity to read this interesting and important study by Dr. Carey and her team. This study is valuable and addresses an important clinical problem for which limited data are available. The authors present a thoughtful analysis and adequately address most of the study limitations. There are a couple of additional comments for their consideration and some suggested analyses that may help address unanswered equations. Major Comments: There is an age-related association seen with older individuals at higher risk of AKI. I wonder whether this is a true effect (i.e. older adults are at increased risk of AKI) or reflects: i) misclassification of CKD as AKI, or ii) imprecision in estimating baseline creatinine in older adults resulting in lower baseline creatinine estimates than expected based on age. One way to ascertain this would be to use the full-age spectrum equation to back-calculate creatinine as the equation will take into consideration changes in GFR across adulthood, particularly in older adults who are expected to have a gradual decline in GFR. Among participants with AKI on admission, what percentage had a decrease of 0.3mg/dL within 48 hours and normal levels at follow-up that might support a definition of AKI rather than misclassification of pre-existing CKD as AKI? Alongside the analysis assessing AKI, I think it would be important to also evaluate the characteristics of participants with CKD at one-month follow-up. Due to the high prevalence of HIV in the adult population, it may not necessarily be appropriate to assume that the premorbid creatinine was normal in PLHIV. Additional sensitivity analysis or adjustment for HIV when assessing CKD risk factors would clarify the likelihood of CKD misclassification as AKI. If this analysis is not possible due to limited sample sizes, a more explicit description of the potential for AKI misclassification would be warranted. I would recommend the authors include estimates of CKD prevalence in PLHIV status across Africa for context. Given the limitations in sample size, the authors may want to consider whether they need to have so many age categories in the logistic regression models, particularly given the wide confidence intervals in the people >70 or whether classification into 20-year categories would be sufficient. Minor comments: In the compiled .pdf there are a few typographical or grammatical errors that need to be addressed on page 2 line 35 "recruited 101 patients recruited" and page 4, line 56 "previously recruitment" The ARK study was just published so the authors could update their manuscript to include the protocol and results on GFR
--

	estimating equations in African populations (https://pubmed.ncbi.nlm.nih.gov/35839814/).
--	---

REVIEWER	Tomlinson, Laurie London School of Hygiene and Tropical Medicine, Epidemiology and Population Health
REVIEW RETURNED	07-Aug-2022

GENERAL COMMENTS	This is a nicely done study on a very important topic which is transparent about the limitations of size and lack of baseline creatinine values. Analytic decisions are appropriate and supported by relevant sensitivity analyses. It would benefit from detailed formatting review. The only major issue I have is some lack of clarity about what the outcome is: the methods state “ Creatinine was measured on admission, at 48-hours and at 3-month follow up. AKI was defined as community acquired if meeting KDIGO criteria on admission. New (incident) AKI was assessed at 48h and diagnosed according to KDIGO using estimated baseline creatinine. AKI recovery was assessed at 48h and defined as recovered if either partial (creatinine within 50% of estimated baseline) or complete (creatinine returned to within 15% of estimated baseline).” So is the AKI being examined in the paper at baseline, at 48h or a composite of both? What if creatinine was high on admission and has gone down since or vice versa? I cannot see any reference being made to the ‘recovery’ data. There is a focus on binary p-values which is a shame given the authors realise the limited power. For example, using bold font for the ‘significant’ associations in table 3 and supplementary tables. I think the table formatting should be reviewed, and the covariates fully labelled (and without needing to refer to methods for definition, e.g. proteinuria) on the left, and I don’t think P-values are useful here where the 95% CI are given. There is no point estimate for diastolic BP in table 3. Is higher or lower diastolic BP associated with AKI? The ordering of the rows could be rationalized for the multivariable model. Given the power limitations and the point that the authors make about Table 2 fallacy I don’t think the Forest plot is meaningful. If the rows show n(%) this should be labelled. In general, acronyms should be spelt out. A histogram of presentation creatinine would be interesting, as would adding BMI and creatinine to supplementary table 2 In Figure 1, how is the HR for HIV death stratified by AKI status? Stratification should give HRs for those with and without AKI. It would be useful to more information about the people with f-u creatinines. What was the reason for missing values in the ~30 people alive without bloods? Just for interest, using a complete-case approach seems to make no meaningful difference to the final model and given the very limited amount of missing data, and how limited multiple imputation would be at this sample size I think the authors might not wish to take that approach next time. Points in the discussion
--

	“Compared with high income settings, where AKI frequently represents renal injury acquired in hospital, our patients had evidence of kidney injury at the point of admission.” I don’t believe this is correct and multiple large epidemiological studies benefitting from baseline creatinines show a large proportion of hospital admissions in HICs have baseline AKI. “However, CKD prevalence of this cohort on presentation is likely to be low given the age demographic” I think this is very unlikely to be true, especially given the high % of people with HIV. As the authors mention, serum creatinine is likely to substantially underestimate CKD prevalence in this community – this is absolutely key for contextualizing the findings. Despite using high-income country based ‘normal’ eGFR values, this study still shows a high proportion of likely baseline AKI – making the findings even more serious. And saying that a substantial proportion may have had baseline CKD on admission also does not undermine the significance of the findings (it just shows that binary AKI/CKD definitions are difficult and may not be meaningful). The ARK study is now published so the authors could update that section and use it to strengthen their discussion on the importance of kidney disease in this community. “GFR estimates are less accurate in the non-steady state such as AKI.” If the authors believe this to be true, why do they report GFR as the first variable in Table 2? Serum creatinine values would be helpful as mentioned above.
--	--

VERSION 1 – AUTHOR RESPONSE

Reviewer: 1

Dr. Rhys Evans, University of Malawi College of Medicine

Comments to the Author:

Thank you for giving me the opportunity to review this manuscript. Carey et al undertake a prospective study to investigate the prevalence and outcomes of sepsis/infection related AKI in a secondary/tertiary hospital in Malawi. They identify risk factors for the development of AKI and AKI outcomes; and they also undertake an exploratory analysis of the use of kidney USS in AKI risk stratification. 101 patients are included in a cohort of young patients with a high prevalence of HIV, typical for hospitals in this region. AKI prevalence is 33% with the majority AKI Stage 3 – similar to previous studies of AKI in Malawi. Mortality at 3 months in the entire cohort was 35% with persistent kidney disease present in 20% of those followed up. AKI development associated with SOFA score, age, and HIV positivity. HIV associated with death.

The authors should be congratulated for undertaking a rigorous study in a challenging resource poor setting. The data complements previous epidemiological studies of AKI undertaken in Malawi and the region. In my mind, its novelty lies in the follow up data collected both in terms of patient and kidney outcomes (expanded on below) and in the exploratory use of kidney USS, albeit the rationale for undertaking this analysis should be clarified, as below.

Comments / suggestions:

1. Whilst some readers of this manuscript will have an insight in to the setting / resources available

where the study was undertaken this is unlikely to be universal. It may be helpful to provide a brief overview of resource limitations (and availability) in the setting where the study was conducted so patient outcomes etc can be interpreted in this light. I.e. what diagnostics /treatments are routinely available. What monitoring / lack therefore. Do patients present directly to the hospital / are they referred etc. How is healthcare set up in Malawi (funding etc).

Thank you for this suggestion. To address this, the Methods section on page 4 includes a *Setting* which now reads:

Zomba Central Hospital (ZCH) is a regional hospital in Southern Malawi serving both urban and rural populations with sixty adult medical inpatient beds. Healthcare is provided free at this government institution at the point of delivery. Approximately half of the patients usually present directly to the hospital and the other half are referred from primary or secondary care. Adults are triaged by a nurse and then reviewed by a doctor or clinical officer. If admission under a specialty team is deemed appropriate, they will be reviewed by an intern or registrar from the admitting speciality, and by a consultant usually within 24 hours.

Limited medical investigations are available on clinician request. Depending on availability, standard of care includes TB diagnostics (sputum smear microscopy, Xpert/MTB/Rif, urine lipoarabinomannan), malaria testing (usually rapid diagnostic test), plain film radiography, full blood counts and serum creatinine ascertainment. Inpatient HIV services include provider-initiated HIV testing and counselling, with CD4 count and viral load available on clinician request. A four bed enhanced observation unit exists on the medical ward with provisions for oxygen therapy, but no cardiac monitoring or defibrillator. There is a 5-bed intensive care unit, mostly used for surgical post operative patients. Local HIV prevalence is estimated at 17% among 15-49 year olds [8].

2. the rationale for the investigation of the use of bedside kidney USS I feel needs expanding. There are no references to previous use of renal USS in the risk stratification of AKI.; for example Is there previous evidence (low or high resource setting) to suggest kidney USS is of use in risk stratifying AKI? Whilst use of USS is routine in the diagnosis of obstruction or alteration in renal arterial / venous flow, I don't feel it is used much beyond this in AKI management as part of standard clinical care.

Thank you for raising this point. Our experience is that ultrasound is clinically important in the assessment of AKI in Malawi due to the limited availability of renal replacement. For decision making regarding transfer of patients to QECH for haemodialysis, renal ultrasound is standard of care to look for evidence of chronic kidney disease. While not an absolute contraindication, acute renal failure without evidence of chronicity would take priority over someone with signs of CKD (e.g. small, echogenic kidneys) for dialysis beds.

We have referred to a systematic review by Ninet et al which reviewed 176 patients with raised RI values on ultrasound, and found they were associated with increased risk of persistent AKI (odds ratio 29.85 (95% confidence interval 8.73-102.16). Given widespread availability of ultrasound in Malawi in comparison to availability of lab creatinine, we felt this was worth investigating further including using the resistive index as a potential predictor of persistent AKI in our cohort. The text now reads:

Kidney ultrasound is standard practice in Malawi as part of the AKI assessment to identify chronic damage in the face of competition for dialysis beds. Patients with evidence of chronic kidney impairment are unlikely to be prioritized for kidney replacement therapy. In addition, kidney ultrasound may be used to identify those at increased risk of AKI, through assessment of kidney morphology, or by doppler estimation of the Renal Resistive Index (RI) which directly correlates with intra-renal arterial resistance.

A systematic review of 176 patients found that raised RI values on ultrasound were associated with increased odds of persistent AKI [7]. Given the expanding availability of ultrasound in Low- and Middle-Income Countries (LMICs), we assessed RI and other sonographic parameters in the diagnosis and risk-stratification of AKI in patients admitted to hospital with infection.

3. Inclusion criteria: the authors state that this is a study of AKI in the setting of 'sepsis', which as in the introduction they state refers to 'life threatening organ dysfunction triggered by a dysregulated host response to infection'. However the inclusion criteria includes patients with 'suspected infection that are admitted to hospital'. Do the authors feel that all of these patients have 'sepsis' and if so it would be useful to outline the definition of sepsis they have used. Would it be more accurate to reframe the study to talk about 'infection related AKI' as opposed to 'sepsis related'. They go on to outline that non-severe infections were excluded. How was the severity of infection determined and by whom?

Thank you for this really useful suggestion. Sepsis as we suggest requires life threatening organ dysfunction, and this was not assessed as part of the inclusion criteria. We have taken your suggestion to refer to our cases as 'infection associated kidney impairment.'

Text now reads:

At ZCH, we performed a prospective observational study recruiting adults (≥ 18 years) with suspected infection and a medical decision to admit to hospital. Suspected infection was deliberately broad to incorporate a wide spectrum of presenting physiology, and to be consistent with ongoing studies. This includes reporting in the last week any of: fever; rigors; night sweats; treatment with antibiotics. All patients in the admissions unit were approached for screening, Monday-Friday, 0700-1700.

We are in agreement that infection related AKI is a closer definition than sepsis-AKI. We have reworded the exclusion criteria referring to non-severe infection which now reads as:

Exclusion criteria were any of: people lacking capacity to consent, with no proxy consent available; pregnancy; non-severe uncomplicated localised infection with only local symptoms apparent on clinical examination by the screening clinical officer (such as cellulitis); detainees; trauma; admission to surgical or obstetric/gynaecological ward; previously recruited to the study.

4. Definitions of AKI: the authors allude to the challenges of AKI diagnosis in resource limited settings in particular given the almost universal lack of historical creatinines through which to determine a baseline value. They use an imputed baseline back calculated based on an assumed GFR of 75 and 100 using MDRD. Did any of the patients have a previous creatinine available or was back calculation used in all patients? As well as creatinines prior to admission, did the authors use / consider use of the lowest creatinine during admission as a baseline value?

In answer to the first part, no participants had a previous creatinine to work on so back calculation was used for all 101 participants. To make this clearer in the text we have amended the wording under AKI assessment in methods:

No participants had a previous documented creatinine result.

We did consider lowest creatinine during admission as a baseline value. The issue we found when using this method combined with our AKI definition (KDIGO $\geq 1.5x$ fold increase over baseline) was exclusion of severe kidney disease at presentation, when there was no change in creatinine over time. For example, if baseline creatinine was 20 mg/dL and remained 20 mg/dL at 48

hours this wouldn't get labelled as acute kidney injury. Perhaps this is appropriate exclusion of potential CKD and avoids misclassification of CKD as AKI.

When we use lowest creatinine as baseline, this identified 17/101 (17%) AKI cases at presentation, which is lower than the other definitions, and has a major impact on the mortality assessment by exclusion of severely impaired function that does not change (new table in supplement), and excluded the most severe cases of kidney disease. Indeed, these cases may actually represent pre-existing CKD rather than AKI, and highlights the limitations of using arbitrary definitions of AKI. We have now reported AKI prevalence using lowest creatinine in Table 2 for comparison with other methods.

Another approach the authors may want to consider in the paper is the concept of acute kidney disease. This has become an increasingly used term that I feel is particularly useful in low resource settings when there is a lack of a baseline value (see Lameire et al, KI, 2021: [https://www.kidney-international.org/article/S0085-2538\(21\)00662-1/fulltext](https://www.kidney-international.org/article/S0085-2538(21)00662-1/fulltext)). AKD can be associated with or without AKI, and allows harmonization with CKD diagnosis. In determining GFR for the purposes of CKD (and hence potentially AKD) definition I note the authors used MDRD - What were reasons for this and was CKD-EPI (race unadjusted) considered? Are there differences in CKD prevalence when different estimating equations are used? What is the prevalence of AKD in the population described – ie. can more patients with kidney disease be identified using this definition?

We have taken the suggestion on board and added AKD to the definition and tables. As a broader definition, which also includes proteinuria and microscopic haematuria, we identified up 63/101 (62%) with evidence of AKD at presentation.

MDRD was chosen over CKD-EPI for the CKD definition because back calculation via MDRD had been used to estimate baseline creatinine, and consistent use of equations was preferred. We have now presented CKD prevalence using both MDRD and CKD-EPI definitions which detect very similar numbers of CKD – 18% and 20%.

5. the authors recruited patients from a 5 month period (jan-july). They may want to comment on malaria seasonality of this period and how this may affect generalizability of results.

Unfortunately, we do not have data from malaria RDTs or blood films, but would have recruited 4/7 across the wetter / "malaria" season. In sepsis admissions in Blantyre, malaria appears 3rd in identified aetiology prevalence (see response to 6.)

6. what are the infections that led to 'sepsis' in the cohort described. How were these infections diagnosed (eg. did all patients with malaria have an film / antigen test)?

Unfortunately, it was not possible to ascertain specific infections, for example through microbiological culture. Other research by Lewis et al suggest in patients with sepsis in Blantyre, 64% (145/225) have a diagnosis identified. Most commonly disseminated mycobacterial infection (76/225 [34%]) followed by culture confirmed bacterial or fungal bloodstream infection (24/225 [11%]) and plasmodium falciparum malaria (21/225 [9%])⁶.

7. Did any of the patients need / receive renal replacement therapy?

Renal replacement therapy is not available in Zomba. Need for renal replacement was not assessed. Standard of care, however, is that patients with acute renal failure (and no evidence of chronic kidney disease) needing dialysis are referred to Queen Elizabeth Central Hospital in Blantyre. Renal

ultrasound is part of this work up to build evidence towards an acute kidney injury versus AKI on chronic kidney disease. AKI on chronic kidney disease is an exclusion for dialysis due to extremely limited capacity.

8. A real strength of the paper I feel is the completeness of the follow up data and the authors should consider expanding more on this. They give the mortality rates at 48 hours and at 3 months. Are there differences in outcomes (mortality) according to kidney disease classification at presentation? If not it may still be useful to present this data.

Thanks for the suggestion, data now presented as a table in the supplement.

Moreover, in the discussion they outline that most deaths occurred post discharge. It may be good to highlight this in the results – ie. what was median LOS – what proportion deaths occurred post discharge. What are the implications for clinical practice of these findings?

Where location of death was known (22/28), 8 deaths were at home following discharge, 14 were in hospital. Unfortunately, discharge date and length of stay were not recorded.

In addition, what was the kidney classification of patients with CKD at 3 months on admission. I assume that the patients with 'ckd' had a GFR <60 on initial presentation in order for the CKD definition to be made. They may want to discuss the possibilities of these patients having undiagnosed CKD (at presentation), or de novo CKD. Did any patients have 'normal kidney function' on presentation and kidney disease at 3 months? – these patients cant be classified as CKD as 3 month duration has not been proven – its unclear if any such patients have been included in the 'ckd group'. Again the authors may want to consider the concept of 'kidney disease' at this timepoint prior to determining whether this is confirmed to be chronic (otherwise acute). Lastly, given 1/5 of patients assessed had kidney disease at the 3 month period it should be emphasized the importance of follow up of this group.

We have defined CKD as an eGFR < 60 mL/min/1.73m² at presentation and 3 month follow up. We were unable to assess for de novo CKD, because as you say, if eGFR > 60 mL/min/1.73m² at presentation, we have not proven the 3 month duration. Of the 7 with CKD at follow up, they were all classified as having AKI and AKD, ie likely misclassifications and made this clear in the limitations discussion. We have kept them in the AKI definition for the reason being that the AKI definitions based on KDIGO are creatinine fold change based, and it is still possible to have AKI on a background of CKD, although in this instance they are likely to represent CKD rather than AKI.

We agree the importance of follow up should be emphasised, if resources allow, or at least focus limited resource on follow up of this group which has been made clearer in the discussion:

Most deaths occurred after discharge rather than in hospital, indicating long term mortality associated with hospitalisation with infection. Combined with the high prevalence of kidney disease identified at 3 months, community follow up of patients is essential. To reduce post-discharge deaths, patients should be counselled on avoiding nephrotoxic medications which are commonly used (for example non-steroidal anti-inflammatories), and advice given on how to prevent future episodes of acute kidney disease. Due to limited resources, follow up should be prioritized for those identified with kidney impairment. Outpatient review after discharge should include assessment for recovery of kidney impairment, new acute kidney disease or formal diagnosis of CKD. Those identified with AKD/CKD will need ongoing follow up for CKD assessment and management to prevent progression to kidney failure.

9. The renal USS analysis I feel is overstated. None of the parameters they refer to reach statistical significance (univariate). It would be interesting to know what the definitions were of: hydronephrosis (21% in AKI and 13% in no AKI seems high to me); loss of CM differentiation; increased echogenicity. A lot of these parameters are somewhat subjective. Was the operator blinded to the kidney diagnosis? The RI is the least subjective in my mind and there are striking similarities in this between the groups (albeit note multivariate findings)

This is a fair comment. We have now removed the statement. The operator was blinded to the kidney diagnosis. Hydronephrosis was defined as dilated appearance of the collecting system. Both have now been stated in the methods.

10. the authors comment that the CKD prevalence is likely to be low. This statement I feel should be made with caution given the risk factors present in the cohort (in particular the prevalence of HIV)

This is another fair comment which we agree with and have removed this statement. It seems plausible that at least 7 have undiagnosed CKD in the cohort.

11. the primary outcomes of the study were to determine the prevalence of AKI and their outcomes. However, in the discussion, there is relatively little comment on these. How did the prevalence and outcomes (kidney / patient) compare to other studies in malawi and the region? How do they compare to high resource settings? Was anything different found in this cohort to what has been previously shown and what might be the reasons for any differences found?

This is a good suggestion, and one which has been developed further in the discussion.

12. As above, how should the finding of high post discharge mortality and development of CKD (/persistent CKD) inform clinical practice in the region. Ie. should patients be monitored more closely post discharge and what are the resource restrictions of this?

This has been further developed in the discussion as highlighted above.

13. the study ref in 23 has recently been published. The authors may want to update and discuss implications of these findings for this study. [https://www.thelancet.com/journals/langlo/article/PIIS2214-109X\(22\)00239-X/fulltext](https://www.thelancet.com/journals/langlo/article/PIIS2214-109X(22)00239-X/fulltext)

This is a great suggestion, and discussion now includes comments on how the ARK study findings impact our study interpretation.

Thank you again and well done again to all the authors for their work so far.

Reviewer: 2

Dr. Andrea Conroy, Indiana University School of Medicine

Comments to the Author:

Thank you for the opportunity to read this interesting and important study by Dr. Carey and her team. This study is valuable and addresses an important clinical problem for which limited data are available. The authors present a thoughtful analysis and adequately address most of the study limitations. There are a couple of additional comments for their consideration and some suggested analyses that may help address unanswered equations.

Major Comments:

There is an age-related association seen with older individuals at higher risk of AKI. I wonder whether this is a true effect (i.e. older adults are at increased risk of AKI) or reflects: i) misclassification of CKD as AKI, or ii) imprecision in estimating baseline creatinine in older adults resulting in lower baseline creatinine estimates than expected based on age. One way to ascertain this would be to use the full-age spectrum equation to back-calculate creatinine as the equation will take into consideration changes in GFR across adulthood, particularly in older adults who are expected to have a gradual decline in GFR.

Thanks for the suggestion, and it raises a good point and so we have now included the FAS equation derived AKI estimates in the supplement to examine the impact of age. The age association appears to remain.

Among participants with AKI on admission, what percentage had a decrease of 0.3mg/dL within 48 hours and normal levels at follow-up that might support a definition of AKI rather than misclassification of pre-existing CKD as AKI?

When looking at either a decrease of 0.3 mg/dL within 48 h OR normal levels of creatinine (< 1.3 mg/dL) at 3 months, 18/33 (55%) with AKI on admission had evidence of recovery, suggesting that these participants had true AKI rather than misclassification. Applying the same application to identify potential misclassification to the AKD participants, 38/63 (60%) had evidence of recovery, with potentially 25/63 (40%) pre-existing CKD misclassified as AKD. Moreover, there are 7 participants with eGFR <60 at both presentation and 3 month follow up who are most likely to represent undiagnosed CKD rather than AKI/AKD.

Alongside the analysis assessing AKI, I think it would be important to also evaluate the characteristics of participants with CKD at one-month follow-up. Due to the high prevalence of HIV in the adult population, it may not necessarily be appropriate to assume that the premorbid creatinine was normal in PLHIV. Additional sensitivity analysis or adjustment for HIV when assessing CKD risk factors would clarify the likelihood of CKD misclassification as AKI. If this analysis is not possible due to limited sample sizes, a more explicit description of the potential for AKI misclassification would be warranted. I would recommend the authors include estimates of CKD prevalence in PLHIV status across Africa for context.

This is a very good point and one we have taken on board. Among the 7 participants with CKD at 3 months, 71% (5/7) were living with HIV. We further agree that it may not be appropriate to assume premorbid creatinine is normal in people living with HIV. In terms of a sensitivity analysis, or adjustment for HIV for assessing CKD risk factors, we feel that the low number (CKD n=7) would limit the interpretation, so have developed the discussion to be more explicit of the potential misclassification of AKI. We have now included:

Estimates for CKD prevalence in people living with HIV across Africa are between 1-46% with a pooled prevalence of 5.6% (95% CI 5.4-5.8)⁷.

We have removed the statement:

CKD prevalence of this cohort on presentation is likely to be low given the age demographic.

Given the limitations in sample size, the authors may want to consider whether they need to have so

many age categories in the logistic regression models, particularly given the wide confidence intervals in the people >70 or whether classification into 20-year categories would be sufficient.

Thank you for this comment. To clarify, we did not discretise age into age categories, but rather included it as a continuous variable using spline terms. This is statistically more efficient than categorisation (mostly through no loss of information inherent in categorisation) and allows us to compare any two ages against each other after model fitting – at no extra cost of statistical power. It is also worth noting that in Table 3, the reference 40 is age 40 not a larger age bracket. The comparisons are age 40 with each of the other ages (20 versus 40, 30 versus 40, 50 versus 40, 60 versus 40, 70 versus 40 and 80 versus 40). We could add a comprehensive comparison of every age against age 40 (or any other age), but chose to just show a few example comparisons.

This is in line with recent guidance from the STRATOS initiative (Topic Group 2 – regression modelling; <https://doi.org/10.1186/s12874-019-0666-3>) as well previous guidance for observational studies (<https://doi.org/10.1097/QAI.0000000000001221>).

We have clarified the methods text and caption to Table 3 to clarify this aspect of the logistic regression model.

Minor comments:

In the compiled .pdf there are a few typographical or grammatical errors that need to be addressed on page 2 line 35 “recruited 101 patients recruited” and page 4, line 56 “previously recruitment”

Thanks for drawing our attention to these errors which have now been corrected.

The ARK study was just published so the authors could update their manuscript to include the protocol and results on GFR estimating equations in African populations (<https://pubmed.ncbi.nlm.nih.gov/35839814/>).

Thank you we have now done this.

Reviewer: 3

Dr. Laurie Tomlinson, London School of Hygiene and Tropical Medicine

Comments to the Author:

This is a nicely done study on a very important topic which is transparent about the limitations of size and lack of baseline creatinine values. Analytic decisions are appropriate and supported by relevant sensitivity analyses. It would benefit from detailed formatting review.

The only major issue I have is some lack of clarity about what the outcome is: the methods state “Creatinine was measured on admission, at 48-hours and at 3-month follow up. AKI was defined as community acquired if meeting KDIGO criteria on admission. New (incident) AKI was assessed at 48h and diagnosed according to KDIGO using estimated baseline creatinine. AKI recovery was assessed at 48h and defined as recovered if either partial (creatinine within 50% of estimated baseline) or complete (creatinine returned to within 15% of estimated baseline).” So is the AKI being examined in the paper at baseline, at 48h or a composite of both? What if creatinine was high on admission and has gone down since or vice versa? I cannot see any reference being made to the ‘recovery’ data.

Thanks for this comment and drawing our attention to the need for greater clarity on our outcomes. We have made the following adjustment to the *AKI assessment* section under methods:

Creatinine was measured on admission, at 48 hours and at 3 month follow up. AKI was defined according to KDIGO as a 1.5 fold increase in baseline creatinine from the estimated baseline [9]. AKI severity as fold increase over estimated baseline was defined according to KDIGO as: stage 1 (1.5-1.9x); stage 2 (2.0-2.9x); stage 3 ($\geq 3.0x$).

We refer to incident AKI at 48 hours if AKI criteria were not met at baseline, but there was a creatinine rise within 48 hours. AKI recovery was defined as a decrease of 0.3 mg/dL between baseline and 48 hours or a normal creatinine (< 1.3 mg/dL) at 3 months.

The recovery data is referred to in table 2 and supplementary figure 3.

We have removed the section titled "Outcomes" and renamed it "Mortality and CKD". Table 2 has been renamed "Outcomes" and presents AKI (by all definitions).

There is a focus on binary p-values which is a shame given the authors realise the limited power. For example, using bold font for the 'significant' associations in table 3 and supplementary tables. I think the table formatting should be reviewed, and the covariates fully labelled (and without needing to refer to methods for definition, e.g. proteinuria) on the left, and I don't think P-values are useful here where the 95% CI are given. There is no point estimate for diastolic BP in table 3. Is higher or lower diastolic BP associated with AKI? The ordering of the rows could be rationalized for the multivariable model. Given the power limitations and the point that the authors make about Table 2 fallacy I don't think the Forest plot is meaningful. If the rows show n(%) this should be labelled. In general, acronyms should be spelt out.

Many thanks for the comment and we fully agree with this. We have reformatted the tables for, we hope, greater clarity. While we did give exact p-values (up to 0.01, below which we simply indicated that the p-value was lower than this), we have now removed the p values altogether (where 95% confidence intervals were given) and removed bold formatting. The formatting of Table 3 has been reviewed. We have removed the Forest plot. The point estimate for diastolic blood pressure has been included. Higher blood pressures were associated with AKI, and we have clarified this in the text and table legend.

A histogram of presentation creatinine would be interesting, as would adding BMI and creatinine to supplementary table 2

Great suggestion, now included.

In Figure 1, how is the HR for HIV death stratified by AKI status? Stratification should give HRs for those with and without AKI.

Thanks for this observation. We have no longer stratified the Cox model by AKI status as the proportional hazard assumption is now met.

It would be useful to more information about the people with f-u creatinines. What was the reason for missing values in the ~30 people alive without bloods?

Yes, many thanks for raising this. Most participants were contacted by phone and asked to come back to hospital for a blood test. For those alive without bloods the reason was usually related to not being able to get to Zomba hospital for logistical reasons related to transport. We have added statement

The main reason for missing follow up data was participants being uncontactable, or unable to get to ZCH for a follow up blood test due to transport or logistical reasons.

Just for interest, using a complete-case approach seems to make no meaningful difference to the final model and given the very limited amount of missing data, and how limited multiple imputation would be at this sample size I think the authors might not wish to take that approach next time.

This is a helpful suggestion, and one which we agree with and for this reason have opted to include only the complete case analysis rather than the imputed data in the revision.

Points in the discussion

“Compared with high income settings, where AKI frequently represents renal injury acquired in hospital, our patients had evidence of kidney injury at the point of admission.” I don’t believe this is correct and multiple large epidemiological studies benefitting from baseline creatinines show a large proportion of hospital admissions in HICs have baseline AKI.

Many thanks for this comment. This statement has now been removed.

“However, CKD prevalence of this cohort on presentation is likely to be low given the age demographic” I think this is very unlikely to be true, especially given the high % of people with HIV. As the authors mention, serum creatinine is likely to substantially underestimate CKD prevalence in this community – this is absolutely key for contextualizing the findings. Despite using high-income country based ‘normal’ eGFR values, this study still shows a high proportion of likely baseline AKI – making the findings even more serious. And saying that a substantial proportion may have had baseline CKD on admission also does not undermine the significance of the findings (it just shows that binary AKI/CKD definitions are difficult and may not be meaningful). The ARK study is now published so the authors could update that section and use it to strengthen their discussion on the importance of kidney disease in this community.

In combination with suggestion from reviewer 2, we have now removed statement:

“However, CKD prevalence of this cohort on presentation is likely to be low given the age demographic”.

We have developed the discussion to keep this in mind and included how the ARK study findings affect interpretation of our study.

“GFR estimates are less accurate in the non-steady state such as AKI.” If the authors believe this to be true, why do they report GFR as the first variable in Table 2? Serum creatinine values would be helpful as mentioned above.

Good point, amended so creatinine appears first.

Reviewer: 1

Competing interests of Reviewer: Nil

Reviewer: 2

Competing interests of Reviewer: I declare no competing interests

Reviewer: 3

Competing interests of Reviewer: None to declare

1. Fabian J, Kalyesubula R, Mkandawire J, et al. Measurement of kidney function in Malawi, South Africa, and Uganda: a multicentre cohort study. *The Lancet Global Health*. 2022;10(8):e11159-e11169. doi:10.1016/S2214-109X(22)00239-X
2. Nakanga WP, Prynne JE, Banda L, et al. Prevalence of impaired renal function among rural and urban populations: findings of a cross-sectional study in Malawi. *Wellcome Open Res*. 2019;4:92. doi:10.12688/wellcomeopenres.15255.1
3. Namazzi R, Batte A, Opoka RO, et al. Acute kidney injury, persistent kidney disease, and post-discharge morbidity and mortality in severe malaria in children: A prospective cohort study. *eClinicalMedicine*. 2022;44. doi:10.1016/j.eclinm.2022.101292
4. Gupta-Wright A, Corbett EL, Oosterhout JJ van, et al. Rapid urine-based screening for tuberculosis in HIV-positive patients admitted to hospital in Africa (STAMP): a pragmatic, multicentre, parallel-group, double-blind, randomised controlled trial. *The Lancet*. 2018;392(10144):292-301. doi:10.1016/S0140-6736(18)31267-4
5. Ninet S, Schnell D, Dewitte A, Zeni F, Meziani F, Darmon M. Doppler-based renal resistive index for prediction of renal dysfunction reversibility: A systematic review and meta-analysis. *J Crit Care*. 2015;30(3):629-635. doi:10.1016/j.jcrc.2015.02.008
6. Lewis JM, Mphasa M, Keyala L, et al. A longitudinal observational study of aetiology and long-term outcomes of sepsis in Malawi revealing the key role of disseminated tuberculosis. *Clin Infect Dis*. Published online August 18, 2021:ciab710. doi:10.1093/cid/ciab710
7. ElHafeez SA, Bolignano D, D'Arrigo G, Dounousi E, Tripepi G, Zoccali C. Prevalence and burden of chronic kidney disease among the general population and high-risk groups in Africa: a systematic review. *BMJ Open*. 2018;8(1):e015069. doi:10.1136/bmjopen-2016-015069
8. Khwaja A. KDIGO clinical practice guidelines for acute kidney injury. *Nephron Clin Pract*. 2012;120(4):c179-184. doi:10.1159/000339789

VERSION 2 – REVIEW

REVIEWER	Evans, Rhys University of Malawi College of Medicine
REVIEW RETURNED	17-Oct-2022
GENERAL COMMENTS	All comments from initial review have been considered and addressed appropriately.
REVIEWER	Conroy, Andrea Indiana University School of Medicine
REVIEW RETURNED	11-Oct-2022
GENERAL COMMENTS	I thank the authors for their responsiveness to the reviewer's comments and edits. This paper is an important contribution to the field and the inclusion of multiple approaches to define baseline creatinine and classify AKI, AKD and CKD provides rich context in a setting where population data on kidney disease are limited.
REVIEWER	Tomlinson, Laurie London School of Hygiene and Tropical Medicine, Epidemiology and Population Health
REVIEW RETURNED	16-Oct-2022
GENERAL COMMENTS	The authors have carefully addressed my concerns and overall this is a valuable and important paper.